# Evaluation of an integrated care pathway for out-of-hospital treatment of older adults with an acute moderate-to-severe lower respiratory tract infection or pneumonia: protocol of a mixed methods study

Rick Roos [1,2] Rianne M C Pepping,[1,2] Maarten O van Aken,[1,2] Geert Labots,[1] Ali Lahdidioui,[1] Johanna M W van den Berg,[3] Nikki E Kolfschoten,[4] Sharif M Pasha,[5] Joris T ten Holder,[6] Susan M Mollink,[7] Frederiek van den Bos,[8] Jojanneke Kant,[9] Ingrid Kroon,[10] Rimke C Vos [2] Mattijs E Numans,[2] Cees van Nieuwkoop [1,2]

For numbered affiliations see end of article.

**Correspondence to**
Rick Roos;
r.roos@hagaziekenhuis.nl

## ABSTRACT

**Introduction** Older adults with an acute moderate-to-severe lower respiratory tract infection (LRTI) or pneumonia are generally treated in hospitals causing risk of iatrogenic harm such as functional decline and delirium. These hospitalisations are often a consequence of poor collaboration between regional care partners, the lack of (acute) diagnostic and treatment possibilities in primary care, and the presence of financial barriers. We will evaluate the implementation of an integrated regional care pathway ('The Hague RTI Care Bridge') developed with the aim to treat and coordinate care for these patients outside the hospital.

**Methods and analysis** This is a prospective mixed methods study. Participants will be older adults (age≥65 years) with an acute moderate-to-severe LRTI or pneumonia treated outside the hospital (care pathway group) versus those treated in the hospital (control group). In addition, patients, their informal caregivers and treating physicians will be asked about their experiences with the care pathway. The primary outcome of this study will be the feasibility of the care pathway, which is defined as the percentage of patients treated outside the hospital, according to the care pathway, whom fully complete their treatment without the need for hospitalisation within 30 days of follow-up. Secondary outcomes include the safety of the care pathway (30-day mortality and occurrence of complications (readmissions, delirium, falls) within 30 days); the satisfaction, usability and acceptance of the care pathway; the total number of days of bedridden status or hospitalisation; sleep quantity and quality; functional outcomes and quality of life.

**Ethics and dissemination** The Medical Research Ethics Committee Leiden The Hague Delft (reference number N22.078) has confirmed that the Medical Research Involving Human Subjects Act does not apply to this study. The results will be published in international peer-reviewed journals.

## STRENGTHS AND LIMITATIONS OF THIS STUDY

⇒ A major strength of this study is that the care pathway has been co-designed by patients from the start, and their interests were the mainstay in the development of the care pathway.
⇒ The satisfaction of patients, their informal caregivers and physicians will be evaluated.
⇒ This study evaluates the real-life application of the care pathway, which ensures that findings can be used immediately to improve the care pathway.
⇒ The mixed methods design of the study enables us to get insight into the feasibility, usability and acceptance of the care pathway, although the sample size of this study will be relatively small for measuring effectiveness and safety.
⇒ The success of the care pathway partially depends on non-medical issues, such as the availability and capacity of homecare institutions and nursing homes.

**Trial registration number** ISRCTN68786381.

## INTRODUCTION

An acute moderate-to-severe lower respiratory tract infection (LRTI) or pneumonia in older adults is generally characterised by diagnostic uncertainty, a high risk of complications and negative outcomes, including mortality.[1 2] Care in the home situation often acutely falls short because of increased dependency due to falls, decline in activities in daily living (ADL) or a state of confusion. This often leads to a presentation at the emergency department (ED) with the goal to define the optimal

treatment plan which usually consists of a combination of antimicrobials, oxygen suppletion and/or inhalation medication, treatment and/or prevention of delirium, and additional help in ADL; and the optimal treatment location. Although, these treatments can be organised outside the hospital, for example, at home or in a nursing home, hospitalisation often occurs because of the 24/7 open access of EDs, and treatment outside the hospital is often considered irresponsible or impossible due to difficulties in ADL and the lack of (available) care.[3–6]

Such hospitalisations of older adults can be considered unnecessary or avoidable when they are related to poor transmural collaboration and different treatment protocols between regional care partners (general practitioners (GPs), hospitals, nursing homes and homecare institutions), the lack of diagnostic and treatment possibilities in primary care, the lack of (acute) availability and capacity in nursing homes and homecare, or the presence of financial barriers.[7–11] Especially in older adults, hospitalisations are associated with iatrogenic harm such as delirium, falls and functional decline.[12–14] As a consequence, older patients often show further decline in ADL from these hospitalisations, and as a result are often transferred to a nursing home or revalidation centre for further recovery. We hypothesise that these hospitalisations may be avoided when the care is well coordinated between care partners.

We, therefore, developed a multidisciplinary regional care pathway 'The Hague RTI Care Bridge', to support GPs with the diagnostics, treatment and organisation of care for older adults with an acute moderate-to-severe LRTI or pneumonia.[15] In this care pathway, clear collaboration agreements were made between involved regional care partners. Three patient journeys were embedded in the care pathway (Figure 1 and Table 1): a hospital-at-home treatment, an ED presentation with priority assessment and admittance to a readily available recovery bed in a nursing home. The care pathway includes a detailed guide on treatment (eg, antibiotics, oxygen suppletion) and its monitoring.[15]

In this prospective mixed methods study, the implementation of the care pathway will be evaluated. During this 12-month study period, it is hypothesised that in at least 75% of the older adults with an acute moderate-to-severe LRTI or pneumonia who are treated outside the hospital according to the care pathway, hospitalisation can be avoided.

### Objectives
#### Primary objective
The primary objective is to determine the feasibility of the care pathway, which is defined as the percentage of patients treated outside the hospital, according to the care pathway, whom fully complete their treatment without the need for hospitalisation within 30 days of follow-up.

#### Secondary objectives
The secondary objectives are to determine the safety of the care pathway (30-day mortality and occurrence of complications (readmissions, delirium, falls) within 30

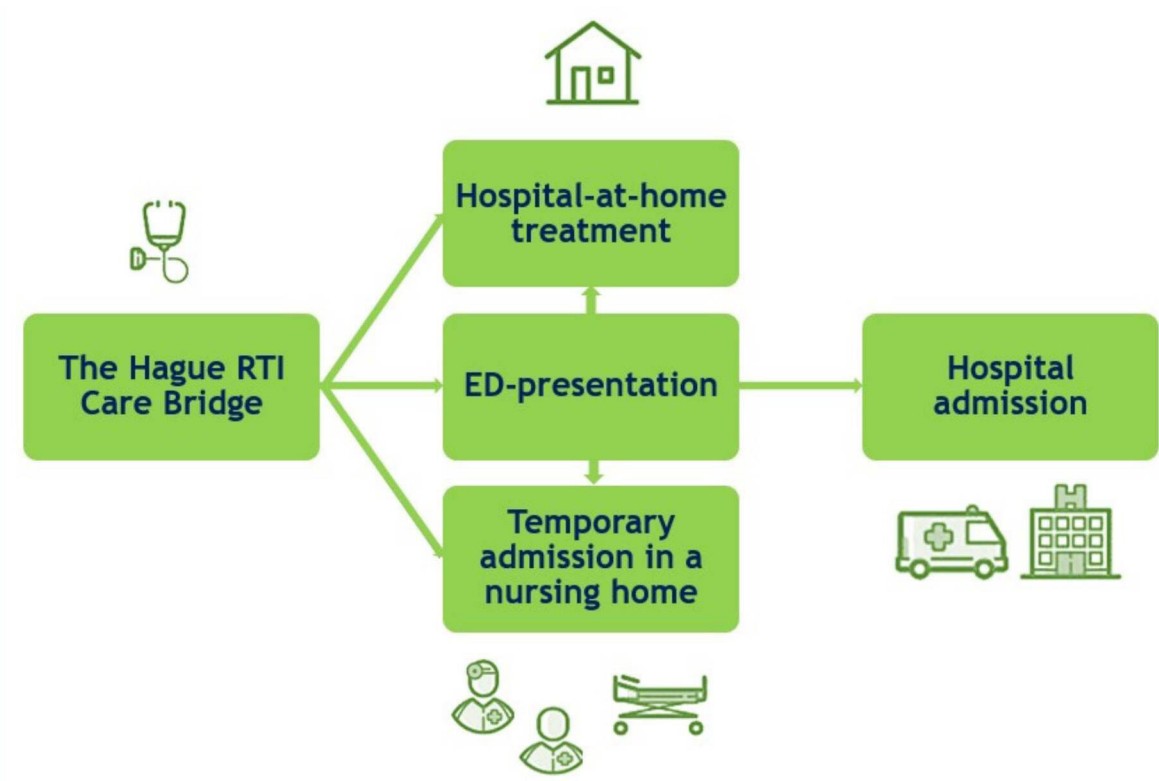

**Figure 1** The three patient journeys in the care pathway. ED, emergency department; RTI, respiratory tract infection.

**Table 1** Overview of the three patient journeys in the care pathway

| | Hospital-at-home treatment | Presentation at the emergency department | Temporary admission in a nursing home |
|---|---|---|---|
| Treatment location | Home | Three options:<br>▶ Home (pathway)<br>▶ Nursing home (pathway)<br>▶ Hospital (regular care) | Nursing home |
| Treatment | Possibilities:<br>▶ Antimicrobials (oral or intramuscular)<br>▶ Oxygen suppletion<br>▶ Inhalation medication<br>▶ Home care | Possibilities:<br>▶ Antimicrobials (oral or intravenous)<br>▶ Oxygen suppletion<br>▶ Inhalation medication<br>▶ Hospital care | Possibilities:<br>▶ Antimicrobials (oral or intramuscular)<br>▶ Oxygen suppletion<br>▶ Inhalation medication<br>▶ Multidisciplinary care |
| Treating physician | General practitioner | Treating physician at ED (and ward when admitted) | Elderly care physician |
| Monitoring | Home monitoring with monitoring kit and registration form (vitals three times a day) | Depends on chosen treatment location* | Monitoring conform the standard of care in the nursing home |

*Hospitalised patients will receive the local standard of care.
ED, emergency department.

days); the satisfaction, usability and acceptance of the care pathway; the total number of days of bedridden status or hospitalisation; sleep quantity and quality; functional outcomes and quality of life (QoL).

## METHODS AND ANALYSIS
### Study design
The design of the study is a prospective mixed methods study, which will be performed in the urban area of The Hague, the Netherlands. In the Netherlands, over 35000 adults are hospitalised with an acute LRTI or pneumonia annually, including 1500–2000 hospitalisations in the area of The Hague. The study period will be from 1 December 2022 to 30 November 2023. In this period, the results will be evaluated frequently for the benefit of interim adjustments.

This is a multicentre study including two teaching hospitals (Haga Teaching Hospital and Haaglanden Medical Center), the largest GP care group and the two biggest homecare institutions of The Hague. The setting will primarily be outside the hospital (primary care or nursing homes). The care pathway offers GPs three options (Figure 1 and Table 1) for the treatment of older adults with a clinical diagnosis of an acute moderate-to-severe LRTI or pneumonia.

### Study population
The study population consists of older adults (age≥65 years) who visit their GP or present at the ED with a clinical diagnosis of an acute moderate-to-severe LRTI or pneumonia. The informal caregivers and treating physicians of participating patients will also be asked to participate in this study to evaluate their experiences and satisfaction about the received or given care.

### Eligibility criteria
To be eligible to participate, a patient must meet all the following criteria: age≥65 years, clinical diagnosis of an acute moderate-to-severe (Pneumonia Severity Index (PSI) class≥3 or Confusion, Urea, Respiratory rate, Blood pressure and age≥65 years (CURB-65) score≥2) LRTI or pneumonia, an oxygen saturation≥92% and a respiratory rate≤24 breaths/minute with a maximum of 5 liters of oxygen (or adjusted oxygen saturation cut-offs as clinically indicated (eg, for patients with chronic obstructive pulmonary disease) by the physician) and written informed consent (IC) for participation.[16 17] Exclusion criteria are: chemotherapy for solid organ malignancy (<2 months before presentation), active haematologic malignancy, immunocompromised status (eg, solid organ transplants) and/or severe dementia (Clinical Dementia Rating Scale Sum Of Boxes score 16–18).[18] To be eligible to participate, an informal caregiver must meet all the following criteria: age≥18 years, being an informal caregiver of a patient included in the study and written IC for participation. To be eligible to participate, a treating physician must meet all the following criteria: physician of a patient included in the study at the main location of treatment, the physician should have treated the patient for at least≥2 (consecutive) days and written IC for participation.

Due to logistical limitations (absence of own GPs and evening/night/weekend shifts in nursing homes), the care pathway will be active on weekdays (Monday–Friday) between 08.00 and 18.00 hours for the hospital-at-home treatment and every day (Monday–Sunday) between 08.00 and 20.00 hours for the admission on a recovery bed in a nursing home. Patients who present at their GP and/or at the ED with an acute LRTI or pneumonia when the care pathway is active will be eligible to be treated according to the care pathway. Patients with an acute LRTI or pneumonia who are hospitalised subsequently to this ED presentation will receive the local standard of hospital care and will therefore not be included.

## Control group

Patients who fulfil the inclusion criteria and do not meet the exclusion criteria of the care pathway, and are hospitalised on weekdays outside office hours (18:00–08:00 hours) or weekend days due to inactivity of the care pathway, will serve as a control group.

## Sample size

Ideally, 50 patients will be treated outside the hospital (care pathway group) and 50 patients will be treated in the hospital (control group). A power analysis was performed for the evaluation of the feasibility of the care pathway. In clinical practice, approximately 10% of the hospitalised patients are readmitted to the hospital after discharge. Our hypothesis is that in at least 75% of the older adults with an acute LRTI or pneumonia who are treated outside the hospital (at home or in a nursing home) according to the care pathway (care pathway group), hospitalisation can be avoided (complete treatment outside the hospital during 30-day follow-up). An 80% power with an alpha of 0.05 for an one-sample study will be achieved if 40 patients are recruited for the care pathway group.

For the qualitative endpoints, data collection (interviews) will be performed until data saturation is reached. When a patient is included and his/her informal caregiver and/or treating physician does not want to participate or cannot participate in the study, the patient will remain in the study and no extra patient will be included to make up for the missing informal caregiver and/or physician.

## Treatment

The treatment of patients outside the hospital will be according to the current Dutch national primary care guidelines.[19–21] In addition to these oral treatment options for bacterial pneumonia (amoxicillin or doxycycline), the treating physician can treat patients in the care pathway with oral moxifloxacin, or intramuscular ceftriaxone in patients with unreliable oral intake or where there is little supervision on the intake, which will be administered by a specialised nurse at home or in the nursing home. Furthermore, the treating physician can treat patients with influenza in the care pathway with baloxavir. Patients who visit the ED but will be treated outside the hospital will receive their first dose of antibiotics intravenously at the ED. The treatment of the control group will be according to local hospital guidelines, which are based on the Dutch national guidelines for bacterial pneumonia (amoxicillin (intravenously or oral) or ceftriaxone (intravenously) for moderate-to-severe pneumonia), COVID-19 and influenza.[22–24] In the care pathway, patients can receive oxygen suppletion with a maximum of 5 liters at home or in the nursing home.

## Monitoring

Patients in the hospital-at-home group will be visited by a specialised nurse within 4 hours after registration. During this visit, the nurse will bring a monitoring kit (pulsoximeter and thermometer) and a printed registration form and will instruct the patient (and his/her informal caregiver) about their use: the patient, informal caregiver and/or nurse will write the vital parameters of the patient down on the registration form at least three times a day. The GP will contact the patient at least once a day to discuss his/her condition and vital parameters together with whether the patient went out of bed that day and whether the patient has fallen. In case of doubt, the GP will visit the patient. Based on these consultations, the GP will monitor the patient, reduce oxygen suppletion if applicable and decide when the monitoring stops. To guarantee a 24/7 safety net, the GP will inform the general practice centre about the patient being treated at home according to the care pathway.

In the nursing home group, monitoring will be similar. Nurses will monitor the vital parameters of the patient at least three times a day and discuss them with the elderly care physician, who will reduce oxygen suppletion if applicable and will decide when the monitoring stops and the patient is fit enough to leave the nursing home. The monitoring of patients in the control group will be according to local hospital standards.

## Outcomes

The primary outcome is the feasibility of the care pathway, which is defined as the percentage of patients treated outside the hospital, according to the care pathway, whom fully complete their treatment without the need for hospitalisation within 30 days of follow-up. This will be measured by the amount of hospitalisations within 30 days of follow-up in patients treated outside the hospital.

The secondary outcomes are the safety of the care pathway (30-day mortality and occurrence of complications (readmissions, delirium, falls) within 30 days); satisfaction, usability and acceptance of the care pathway (30-day satisfaction questionnaires (patients, informal caregivers and treating physicians) and semi-structured in-depth interviews with the first 10 patients and their informal caregivers in the hospital-at-home group (and the treating GPs), nursing home group and control group after 2–3 weeks); total number of days of bedridden status or hospitalisation; sleep quantity in the first 2 days after inclusion and on the 7th day as assessed by the core Consensus Sleep Diary; sleep quality, functional outcomes and QoL.[25] Sleep quality will be assessed by the Patient-Reported Outcomes Measurement Information System (PROMIS) Sleep Disturbance short form 8b on day 7 and day 30 after inclusion.[26 27] Functional outcomes will be measured using KATZ-15 at 30 days, 6 and 12 months.[28–30] QoL will be assessed using EQ-5D-5L at 30 days, 6 and 12 months.[31 32] If available, Dutch validated questionnaires are used.

**Table 2** Diagnostics packages at the primary care centre and the emergency department

| | Primary care centre | Emergency department |
|---|---|---|
| Laboratory tests | Optional | ► Blood cell count including differentiation<br>► C reactive protein<br>► Sodium/potassium/glucose<br>► Kidney function (creatinine/urea)<br>► Optional: D-dimer/NT-proBNP |
| Microbiology | Nasopharyngeal swab (PCR):<br>► SARS-CoV-2<br>► Influenza A/B<br>► Respiratory syncytial virus | Nasopharyngeal swab (PCR):<br>► SARS-CoV-2<br>► Influenza A/B<br>► Respiratory syncytial virus |
| Radiology | Optional | X-ray and/or CT scan of the thorax |
| Clinical prediction rules | ► Adjusted APOP screening | ► APOP screening<br>► PSI or CURB-65 measurement |
| Electrocardiogram | Optional | Yes |

APOP, Acute Presenting Older Patient; CURB-65, Confusion, Urea, Respiratory rate, Blood pressure and age≥65 years; NT-proBNP, N-terminal pro brain natriuretic peptide; PCR, Polymerase Chain Reaction; PSI, Pneumonia Severity Index; SARS-Cov-2, Severe Acute Respiratory Syndrome Coronavirus 2.

## Study procedures

### Patient presenting at GP

When a GP decides to treat a patient at home or in a nursing home according to the care pathway, the GP will perform a physical examination, measure the vital parameters (heart rate, blood pressure, respiratory rate, oxygen saturation and temperature) and will perform the standard diagnostics package (Table 2): nasopharyngeal swab (SARS-CoV-2, influenza and respiratory syncytial virus (RSV)) and the adjusted Acute Presenting Older Patient (APOP) screening.[33] The GP will inform the patient (or representative (eg, in case of incapacity due to dementia/delirium)) about the study on inclusion in the care pathway and asks for oral IC. When the patient or representative (on behalf of the patient) agrees to participate, the GP will hand over the patient information leaflet (PIL) and inform the research team. Within one workday, a research team member visits the patient (and representative if applicable) at home or in the nursing home to provide written IC and collect data.

When a patient does not want to participate in the study, no research data will be collected and the patient will still be managed according to the care pathway.

### Patient presenting at the ED

When the GP decides to refer the patient to the ED for additional assessment or a patient presents at the ED, a predefined diagnostics package (Table 2) will be performed including laboratory tests, nasopharyngeal swab (SARS-CoV-2, influenza and RSV), clinical prediction rules (PSI or CURB-65, and APOP), imaging of the chest (X-ray and/or CT scan) and an electrocardiogram. The treating ED physician will inform the patient (and representative if applicable) about the study (including handing over the PIL) on inclusion in the care pathway (hospital-at-home treatment or nursing home admission) or on hospitalisation for patients eligible for the control group, and will ask the patient (or representative if applicable) for oral IC to participate. Within one workday, a research team member visits the patient (and representative if applicable) on location to provide written IC and collect data.

### Monitoring data

Besides the data from the registration forms of patients in the hospital-at-home group, GPs will collect information regarding illness duration during their treatment/monitoring of the patient, thereby providing insight into the occurrence of complications and the percentage of complete treatments at home. Elderly care physicians will collect similar information in the electronic health records (EHRs) of patients in the nursing home group. For patients in the control group, this information will be extracted from the hospital EHRs.

### First study visit

A research team member will visit the patient on location (at home, in the nursing home or hospital) on the first workday after the start of the treatment. During this visit, patients (and their representatives if applicable) will be able to ask additional questions about the PIL and the study. If a patient (or representative on behalf of the patient) agrees to participate, written IC will be asked for participation. In case a representative has provided written IC for an incapacitated patient (eg, in case of delirium), and the patient's medical condition improves over time, the patient will be asked for written IC when the patient is considered competent again. When a patient has given written IC, their informal caregiver and treating physician will also be approached to participate in the study.

During this first visit, baseline information will be collected from the patient. This will include demographic information and a geriatric assessment (GA). A GA is an evidence-based, systematic procedure used to objectively describe the health status of older adults, focusing on somatic, functional and psychosocial domains; and aimed at constructing a multidisciplinary treatment plan. The GA will include the following validated tests: the Charlson Comorbidity Index, G-8 screening tool, 6-item Cognitive

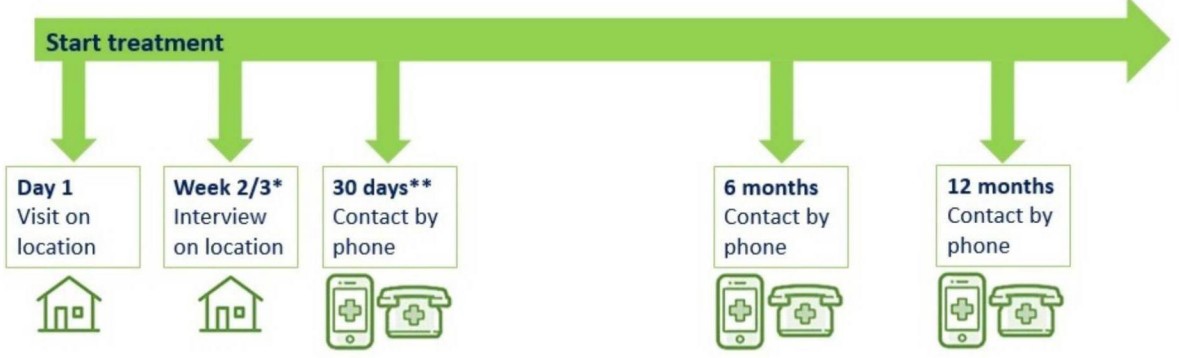

**Figure 2** Overview of preferable study contact moments for patients. *The interview will be held with the first 10 patients and informal caregivers in the hospital-at-home group (and general practitioners), the nursing home group and the control group. **Informal caregivers and treating physicians will also receive a phone call at 30 days.

Impairment Test, functional status (KATZ-15 and living situation) and QoL (EQ-5D-5L).[34–36] Ethnicity and religion are included in the demographic information as the area of The Hague has a multicultural society and these factors may influence a patient's care system and thereby the choice of treatment location. Dementia research has shown that a considerable amount of people with a migration background makes limited use of professional homecare, and needed care is often taken over by family members.[37 38]

During this visit, patients will receive a core Consensus Sleep Diary to fill in on the 2 upcoming days and on the 7th day after inclusion. Patients will also receive a PROMIS Sleep Disturbance short form 8b to fill in on the 7th day after inclusion. The sleep quality/quantity forms will be collected 1 week after inclusion. In the hospital-at-home group, the registration forms will be collected together with the monitoring kits when the patient is released from the care pathway. See Figure 2 and Table 3 for an overview of the time points and collected data.

### Semi-structured interview after 2–3 weeks

After 2–3 weeks of inclusion, a semi-structured in-depth interview will be held on voluntary basis with the first 10 patients included in the hospital-at-home group, nursing home group and control group. If these 10 patients in each group agree to participate in the interview, their informal caregivers will also be asked if they want to participate in a similar interview to collect information about their experiences. The interview of the patient and their informal caregiver can take place simultaneously. If the patients in the hospital-at-home group agree to participate in the interview, their GPs will also be asked for an interview to collect information about their experiences with the care pathway. This interview with the GP will take place separately. By selecting participants in this way, the interviews will be taken without purposive sampling.

The framework that is used to develop the interview guide is the Consolidated Framework for Implementation Research (https://cfirguide.org), which provides a

| Table 3 | Overview of the data collection at the different time points | | | | | | |
|---|---|---|---|---|---|---|---|
| | Day 1 | Day 2 | Day 7 | Weeks 2–3 | 30 days | 6 months | 12 months |
| Mortality | X | X | X | | X | X | X |
| KATZ-15 | X | | | | X | X | X |
| Living situation | X | | | | X | X | X |
| EQ-5D-5L | X | | | | X | X | X |
| Demographics | X | | | | | | |
| Charlson Comorbidity Index | X | | | | | | |
| G-8 screening tool | X | | | | | | |
| 6-item Cognitive Impairment Test | X | | | | | | |
| Core Consensus Sleep Diary | X | X | X | | | | |
| PROMIS Sleep Disturbance short form 8b | | | X | | X | | |
| Satisfaction | | | | X* | X† | | |
| Complications | | | | | X | | |

*An interview will be held with the first 10 patients and informal caregivers in the hospital-at-home group (and general practitioners), the nursing home group and the control group.
†Questionnaires will be conducted with the patients, their informal caregivers and their treating physicians.
PROMIS, Patient-Reported Outcomes Measurement Information System.

framework of constructs that are associated with effective implementation.[39–41] There are five domains with corresponding example questions. These questions have been adapted and tailored to the intervention programme and will form the basis for a process evaluation of the implementation of the care pathway focusing on the implementation, the mechanisms of impact and context (facilitators/barriers to implementation).[42] The information collected during these interviews will be used to adjust the care pathway.

### Study phone calls at 30 days

At 30 days, all patients will receive a phone call (or visit at request of the patient) from a research team member in which they will be asked about the occurrence of complications (readmissions, delirium, falls), their sleep quality (PROMIS Sleep Disturbance short form 8b), functional status (KATZ-15 and living situation), QoL (EQ-5D-5L) and satisfaction. The questions to evaluate satisfaction are based on the Consumer Quality Index, Patient Reported Outcome Measures and other research evaluating the out-of-hospital treatment of patients and adjusted if applicable.[43 44]

At 30 days, all informal caregivers and treating physicians will receive a phone call from a research team member in which they will be asked about the satisfaction, usability and acceptance of the care pathway. In this call, treating physicians will also be asked about the occurrence of complications.

### Study phone call at 6 and 12 months

At 6 and 12 months, all patients will receive a phone call (or visit at request of the patient) from a research team member in which they will be asked about their functional status (KATZ-15 and living situation) and QoL (EQ-5D-5L).

The research group has longstanding experience performing questionnaires by telephone in the older population, which has proven to be feasible and was validated in previous studies.[45]

### Data analysis plan

The primary outcome is the feasibility of the care pathway. This will be quantitative data. Categorical variables will be presented as counts and frequencies. The quantitative data will be analysed with the Statistical Package for the Social Sciences (SPSS) V.28.

The secondary outcomes will be established by comparing the patients in the care pathway group with the patients in the control group during follow-up. Secondary outcomes are the safety of the care pathway (30-day mortality and occurrence of complications within 30 days); the satisfaction, usability and acceptance of the care pathway; the total number of days of bedridden status or hospitalisation; sleep quantity and quality; functional outcomes and QoL.

Categorical variables will be presented as counts and frequencies. Differences between groups will be tested with $\chi^2$ tests and multivariate logistic regression models. Continuous data will be presented as means (standard deviations) for normally distributed data or medians (interquartile ranges) for non-normally distributed data. Differences between groups will be tested with independent t-tests or one-way Analysis Of Variance (ANOVA) tests (normal distribution), or Mann-Whitney U or Kruskal-Wallis tests (no normal distribution) depending on the amount of groups to be compared per analysis, and by multivariate linear regression models. All tests of significance will be at two-tailed 0.05 level. The 95% confidence intervals will be used to assess the presence/absence of associations. The quantitative data will be analysed with SPSS V.28.

All recorded interviews will be transcribed by two research team members and hereafter then coded with Atlas.ti V.22. The interview recordings will be saved in a secured folder on the network of the coordinating hospital and will be deleted after verbatim transcription. Transcriptions will be saved on the network of the coordinating hospital. We will apply thematic content analysis to identify and categorise recurrent themes/elements in the interviews. Thereby, we aim to classify the qualitative data in the right sections (implementation, methods, context) of the process evaluation of the implementation of the care pathway.

### Patient and public involvement

This care pathway has been co-designed by patients from the start and their interests were the mainstay in the development of the care pathway. Patients will also be involved in the evaluation of the care pathway. After every time, five patients have been treated at home according to the care pathway, a group of stakeholders (including patient and public representatives) will evaluate the experiences and outcomes of care, and decide whether to continue, adjust or stop the use of the care pathway.

### DISCUSSION

This study will evaluate a care pathway for the treatment of older adults with an acute moderate-to-severe LRTI or pneumonia with the aim to treat patients outside the hospital. The results of this study will provide evidence whether treatment at home or in a nursing home is feasible, usable and satisfactory for patients, their informal caregivers and treating physicians. Hospitalisation of older adults with a LRTI or pneumonia is often related to poor collaboration and different treatment protocols between regional care partners (GPs, hospitals, nursing homes and homecare institutions), the lack of diagnostic and treatment possibilities in primary care, the lack of availability and capacity in nursing homes and homecare and the presence of financial barriers.[7–11] These hospitalisations may be avoided by good collaboration between regional care partners and shared treatment and management protocols.

The prospective observational study by Marrie *et al* showed that a substantial number of patients in PSI risk classes IV–V could be safely treated at home.[6] However, to our knowledge, no studies have been performed that focus on the treatment of older adults with an acute moderate-to-severe LRTI or pneumonia outside the hospital. Therefore, we aim to get insight into the outpatient treatment of older adults with an acute moderate-to-severe LRTI or pneumonia. A major strength of this study is that the care pathway has been co-designed by patients from the start. Their interests were the mainstay in the development of the care pathway, and the satisfaction of the patients, their informal caregivers and physicians will be evaluated and used to improve the care pathway. Besides that, patients will play a key role in the periodic evaluations in which experiences and outcomes of care will be multidisciplinary evaluated. The evaluation of the real-life application of this care pathway ensures that the findings can be used immediately to improve the care pathway.

Despite its strengths, this study does have some limitations. First, the success of the care pathway partially depends on non-medical issues, such as the availability and capacity of nursing homes and homecare. During the course of this study, these barriers will be evaluated and adjustments will be made if necessary. Another limitation is that the allocation between the care pathway group and the hospital group is based on the time of day and day of the week. Research has shown that illness severity is generally greater outside office hours.[46] Though, we aim to limit this bias by using limits for oxygen saturation, respiratory rate and maximum of oxygen suppletion in the inclusion criteria, the data of clinical outcomes should therefore be interpreted with caution and be considered explorative rather than proof. Furthermore, it will be challenging to keep all healthcare professionals informed as the care pathway is implemented in an urban area where many healthcare professionals are active. The mixed methods design of the study enables us to get insight in the feasibility, usability and acceptance of the care pathway, although the sample size will be relatively small for measuring effectiveness.

In summary, The Hague RTI Care Bridge will improve our understanding of the possibility and the feasibility to treat patients with an acute moderate-to-severe LRTI or pneumonia outside the hospital. In addition, it will give insight in this new cooperation between two teaching hospitals, the largest GP care group and the two biggest homecare institutions of The Hague, thereby creating possibilities for a similar outpatient treatment for patients with other medical problems (eg, erysipelas and urinary tract infections).

## ETHICS AND DISSEMINATION
### Ethics
The Medical Research Ethics Committee Leiden The Hague Delft (reference number: N22.078) has confirmed that the Medical Research Involving Human Subjects Act does not apply to this study. The Haga Teaching Hospital Institutional Scientific Review Board approved this study (reference number: T22-066). This study is registered at ISRCTN (reference number: ISRCTN68786381).

### Dissemination
All relevant results will be disseminated through publications in international peer-reviewed journals and presentations at scientific conferences. No identifiable patient data will be disseminated.

### Data availability
The datasets, including the coded participant-level data, will be made available to other researchers on reasonable request after the publication of the study results. Requests should be directed to the coordinating investigator RR. Data requestors will need to sign a data access agreement. These datasets will only contain the coded individual-level data that underlie the results of the publication the researcher is referring to in his/her request. Participants will be asked for their consent to share their coded data on reasonable request with researchers in other countries.

**Author affiliations**
[1]Department of Internal Medicine, Haga Teaching Hospital, The Hague, The Netherlands
[2]Health Campus The Hague/Department of Public Health and Primary Care, Leiden University Medical Center, The Hague, The Netherlands
[3]Department of Pulmonology, Haga Teaching Hospital, The Hague, The Netherlands
[4]Department of Emergency Medicine, Haga Teaching Hospital, The Hague, The Netherlands
[5]Department of Internal Medicine, Haaglanden Medical Center, The Hague, The Netherlands
[6]Department of Pulmonology, Haaglanden Medical Center, The Hague, The Netherlands
[7]Department of Emergency Medicine, Haaglanden Medical Center, The Hague, The Netherlands
[8]Department of Internal Medicine, Leiden University Medical Center, Leiden, The Netherlands
[9]Hadoks, The Hague, The Netherlands
[10]Kroon Elderly Care Physician, The Hague, The Netherlands

**Contributors** The protocol was drafted by RR, RMCP and CvN and was refined by all authors (MOvA, GL, AL, JMWvdB, NEK, SMP, JTtH, SMM, FvdB, JK, IK, RCV and MEN). RR and RMCP were responsible for drafting the manuscript and contributed equally. All other authors (MOvA, GL, AL, JMWvdB, NEK, SMP, JTtH, SMM, FvdB, JK, IK, RCV, MEN and CvN) contributed to the manuscript and read and approved the final manuscript.

**Funding** This work was supported by the Netherlands Organisation for Health Research and Development (ZonMw) grant number 1010003222003, MenzisFonds grant number M2022-W004, Dr. C.J. Vaillantfonds reference number 20220809, Bavo stichting grant number 20220024 and Wetenschapsfonds HagaZiekenhuis grant number T22-066.

**Competing interests** None declared.

**Patient and public involvement** Patients and/or the public were involved in the design, or conduct, or reporting, or dissemination plans of this research. Refer to the Methods and analysis section for further details.

**Patient consent for publication** Not applicable.

**Provenance and peer review** Not commissioned; externally peer reviewed.

**ORCID iDs**
Rick Roos http://orcid.org/0000-0002-7611-5301
Rimke C Vos http://orcid.org/0000-0003-1074-6255
Cees van Nieuwkoop http://orcid.org/0000-0003-0734-0844

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
