## [Reviewer comments · BMJ Open]

ARTICLE DETAILS

TITLE (PROVISIONAL)	Evaluation of an integrated care pathway for out-of-hospital treatment of older adults with an acute moderate-to-severe lower respiratory tract infection or pneumonia: protocol of a mixed methods study
AUTHORS	Roos, Rick; Pepping, Rianne M.C.; van Aken, Maarten O.; Labots, Geert; Lahdidioui, Ali; van den Berg, Johanna; Kolfschoten, Nikki E.; Pasha, Sharif M.; ten Holder, Joris T.; Mollink, Susan M.; van den Bos, Frederiek; Kant, Jojanneke; Kroon, Ingrid; Vos, Rimke C; Numans, Mattijs; van Nieuwkoop, Cees

VERSION 1 – REVIEW

REVIEWER	Gulliford, Martin King's College London, UK
REVIEW RETURNED	24-Mar-2023

GENERAL COMMENTS	This paper provides a careful and thorough description of an evaluation of a care pathway for respiratory infection. In most respects the paper seems ready for publication. However, it was not clear whether this study represents research or service evaluation. While research will produce generalisable knowledge, service evaluation provides information that is mainly useful in the local context. For example, see https://www.hra-decisiontools.org.uk/research/docs/DefiningResearchTable_Oct2017-1.pdf Aspects of the study that appeared more like service evaluation: - the intervention is only available at a single site.- the intervention is not well described (reference 15 was not accessible).- there is no theory or logic model to support this intervention. If it 'works' it may not be clear what is working.- this is a non-randomised study and necessarily at high risk of bias. Basing allocations on time of day or day of the week, may be especially troublesome as it is well documented that severity of illness is generally greater 'out of hours'. These issues merit debating in the paper. This is a mixed methods study but the qualitative component could be better theorised. What questions will the qualitative research aim to answer? There needs to be a process evaluation to understand the contextual factors associated with implementation outcomes.
--

REVIEWER	Sibomana, Jean-Pierre Centre Hospitalier Universitaire de Butare, Medicine
-----------------	---

REVIEW RETURNED	14-Apr-2023
GENERAL COMMENTS	The study is very interesting but some issues should be clear for patient safety. In follow up first 7 days should be followed daily. Second, though place of care will be different, care should be equal and protocolized with similarities for safety of patients. Cost saving and logistical impact method be clear rather than giving the ambiguity you may omit this second object if you are not sure that you will do it. For safety reason, please mention how you will be preventing the worst outcome in care pathway.

VERSION 1 – AUTHOR RESPONSE

Comments from reviewer 1 (Dr. Martin Gulliford, King’s College London, UK) (26-May-2023)

Comment: This paper provides a careful and thorough description of an evaluation of a care pathway for respiratory infection. In most respects the paper seems ready for publication. However, it was not clear whether this study represents research or service evaluation. While research will produce generalizable knowledge, service evaluation provides information that is mainly useful in the local context. For example see: https://www.hra-decisiontools.org.uk/research/docs/DefiningResearchTable_Oct2017-1.pdf

Response: Many thanks for this critical point and the useful suggestion of the decision tool. Though, our study represents a process evaluation of the implementation of a developed care pathway, we judge it as research producing generalizable knowledge. One of the main aims of our study is to gather knowledge on the experiences of patients, their informal caregivers and treating physicians whether treatment outside the hospital (at home or in a nursing home) according to the care pathway is satisfactory, feasible, usable, and acceptable. As such, this study will give insights and generalizable knowledge in whether older adults with a moderate-to-severe LRTI or pneumonia are willing to be treated outside the hospital, and whether the use of a similar care pathway is possible elsewhere.

Besides that, the mixed methods design provides the possibility to evaluate clinical outcomes. According to current (inter)national guidelines, moderate-to-severe LRTI or pneumonia should be treated empirically with intravenous antibiotics and thus patients need to be hospitalised. This study can be considered as a proof-of-principle study regarding the effectiveness and safety of treating moderate-to-severe LRTI or pneumonia outside the hospital with oral (or intramuscular) antibiotics. Thereby, this study will lead to generalizable knowledge on the treatment of moderate-to-severe LRTI or pneumonia outside the hospital.

Comment: Aspects of the study that appeared more like service evaluation: - the intervention is only available at a single site.

Response: The care pathway is active in the entire urban area of The Hague, one of the biggest cities in the Netherlands, including many primary care centers, hospitals, and nursing home. We have changed the study design section of the methods and analysis section, highlighting that it is a multicentre study including the two teaching hospitals (Haga Teaching Hospital and Haaglanden Medical Center), two nursing homes and the primary care centers in the urban area of The Hague.

Comment: Aspects of the study that appeared more like service evaluation: - the intervention is not well described (reference 15 was not accessible).

Response: The intervention is the implementation of the care pathway into clinical practice. As such, patients will be treated accordingly and preferably at home with oral (or intramuscular) antibiotics instead of being hospitalised and treatment with intravenous antibiotics. The development of the integrated care pathway has been described in a separate manuscript, that recently has been accepted for publication in the International Journal of Integrated Care. For the reviewers and editors interest, we send you a copy of this paper. It is anticipated that it will be published in June or July 2023 and hopefully the official reference can be added in this current paper during the editing and styling process, once it has been accepted for publication in BMJ Open. For now, we have changed reference 15 into: Pepping RMC, van Aken MO, Vos RC, et al. Using design thinking for co-creating an integrated care pathway including hospital at home for older adults with an acute moderate-severe respiratory infection in the Netherlands, *International Journal of Integrated Care*. 2023 (in press).

Comment: Aspects of the study that appeared more like service evaluation: - there is no theory or logical model to support this intervention. If it 'works' it may not be clear what is working.

Response: The theory behind this intervention (the implementation of the care pathway) is that hospitalisation is not always necessary for the treatment of patients with moderate-to-severe pneumonia as stated by Marrie et al. (reference 6). It is generally known that older adults who are hospitalised have a greater risk of complications such as delirium and falls. Besides that, older adults often do not want to be hospitalised. These principles form the rationale behind the development and implementation care pathway, namely, to prevent unnecessary admissions in older adults who present at the emergency department with a LRTI or pneumonia. The process evaluation of the implementation of the care pathway through the interviews with the patients, informal caregivers and general practitioners will give insight in whether and why the care pathway is working or not. Besides that, these will also give insight in barriers and facilitators in the implementation of the care pathway.

Comment: Aspect of the study that appeared more like service evaluation: - this is a non-randomised study and necessarily at high risk of bias. Basing allocations on time of day or day of the week, may be especially troublesome as it is well documented that severity of illness is generally greater 'out of hours'. These issues merit debating in the paper.

Response: We agree with the reviewer that data on the clinical outcomes must be interpreted with caution and should be considered explorative. We have changed the discussion section accordingly. This limitation of basing our allocation on time of day or day of the week is added to the limitations part of the discussion of the paper, including reference 40. In our study, we aim to limit this bias by using limits for oxygen saturation and respiratory rate, and a maximum liters of oxygen supply in the inclusion criteria.

Comment: This is a mixed methods study but the qualitative component could be better theorised. What questions will the qualitative research aim to answer? There needs to be a process evaluation to understand the contextual factors associated with implementation outcomes.

Response: As noted above, one of the main aims of our study is to gather knowledge on the experiences of patients, their informal caregivers and treating physicians whether treatment outside the hospital (at home or in a nursing home) according to the care pathway is satisfactory, feasible, usable, and acceptable. This is crucial in our process evaluation of the implementation of the care pathway focusing on the implementation, mechanisms of impact and context (facilitators/barriers to implementation), see also the added reference 36. We added this theory to the qualitative analysis in the protocol paper. This process evaluation will support the interpretation of the quantitative results.

Comments from reviewer 2 (Dr. Jean-Pierre Sibomana, Centre Hospitalier Universitaire de Butare) (26-May-2023)

Comment: This study is very interesting but some issues should be clear for patient safety. In follow up first 7 days should be followed daily.

Response: All patients will be followed up by a physician as long as clinically indicated by the treating physician. For patients treated at home, patients will be visited by homecare nurses three times a day for monitoring as long as clinically indicated by the general practitioner. For patients treated at a nursing home, the elderly care physician will decide when the patient is able to go back to his/her own living environment. For patients treated at the hospital, the treating physician at the ward will decide when the patient is able to leave the hospital (to go home with/without homecare, nursing home). This monitoring process is also well documented in reference 15. In the study, patients will be visited by a member of the research team on day 1 and on day 7 when the sleep forms are collected.

Comment: Second, though place of care will be different, care should be equal and protocolized with similarities for safety of patients.

Response: In the care pathway, clear agreements are made on which antibiotics, antivirals, steroids and inhalation medication can be used in the treatment of patients outside the hospital. In the hospitals, the standard of care will be given. In the nursing homes, nurses will be present 24/7 to monitor the patient his/her condition similar as in the hospitals. At home, homecare nurses will visit the patient if required at least three times a day to monitor his/her condition. However, to make sure patients treated at home receive help when required, patients will be able to contact the contact centre from the homecare institution 24/7. At this centre, a triage nurse will decide what is necessary. More details on the organisation of care in the care pathway can be found in reference 15 (Pepping et al.)

Comment: Cost saving and logistical impact method be clear rather than giving the ambiguity you may omit this second object if you are not sure that you will do it.

Response: We have removed cost savings and logistical impact as secondary objectives/outcomes as both are not our main focus of the article and it is still uncertain whether it will be possible to do these analyses. Therefore, these objectives have been removed from the entire paper.

Comment: For safety reason, please mention how you will be preventing the worst outcome in care pathway.

Response: In the nursing homes, nurses will be present 24/7 to monitor the patient his/her condition. At home, homecare nurses will visit the patient three times a day to monitor his/her vital parameters and clinical condition. However, to make sure patients treated at home receive help when required, patients will be able to contact the contact centre from the homecare institution 24/7. At this centre, a triage nurse will decide what is necessary (e.g. an additional check from a homecare nurse, a visit from a general practitioner, an ambulance) for the patient at that moment. The general practitioner will contact the patient and/or his informal caregiver by phone or a home visit once a day to monitor the patient his/her collected vital parameters and the clinical condition of the patient. In addition, the general practitioner will notify the general practitioner urgency centers about the hospital-at-home treatment of the patient, thereby making sure that the general practitioners who are sitting on the general practitioner urgency centers outside office hours are aware of the treatment of the patient providing a 24/7 safety network. Besides that, the elderly care physicians and general practitioners are always able to contact the pulmonologists/internists in the hospitals and refer a patient to the emergency department for additional assessment or taking over the treatment.